# FINGERTIP 20K: A BENCHMARK FOR PROACTIVE AND PERSONALIZED MOBILE LLM AGENTS

**Qinglong Yang, Haoming Li, Haotian Zhao, Xiaokai Yan, Jingtao Ding, Fengli Xu,\* Yong Li\***
Department of Electronic Engineering
Tsinghua University
Beijing, China

## ABSTRACT

Mobile GUI agents are becoming critical tools to improve user experience on smart devices, with multimodal large language models (MLLMs) emerging as the dominant paradigms in this domain. Current agents, however, rely on explicit human instructions, overlooking the potential to leverage the contextual information (like location, time, user profile) and historical data for proactive task suggestions. Besides, previous works focus on optimizing the success rate during task execution, but pay less attention to the personalized execution trajectory, thereby neglecting potentially vast differences in user preferences. To address these challenges, we introduce the FingerTip 20K benchmark. We collected 20K unique human demonstrations of multi-step Android device interactions across a variety of everyday apps. These demonstrations are not isolated but are continuously acquired from the users' long-term usage in their real lives, and encompass essential user-related contextual information. The benchmark contains two new tracks: proactive task suggestions by analyzing environment observation and users' previous intents, and personalized task execution by catering to users' action preferences. Our experiments reveal that the tracks we propose pose significant challenges for leveraging user-related information in GUI tasks. We also performed a human study to show that there exists a huge gap between existing agents and humans. The model fine-tuned with the data we collected effectively utilized user information and achieved good results, highlighting the potential of our approach in building more user-oriented mobile LLM agents. Our code is open-source at `https://github.com/tsinghua-fib-lab/FingerTip-20K` for reproducibility.

## 1 INTRODUCTION

Recent studies have explored how to utilize multimodal large language models (MLLMs) to build graphical user interface (GUI) control agents (Koh et al., 2024; Zheng et al., 2024; Yan et al., 2023; Kim et al., 2023; Deng et al., 2023), with a significant direction being mobile phone GUI control agents. These mobile LLM agents have the potential to tremendously improve user experience with mobile devices, since GUI is a universal interface across various applications. These agents receive a natural language task instruction, such as "Set an alarm for 7:30 for me", and then perceive the device state by observing the device screen (via screenshots or textual UI trees), and generate actions (click, type, scroll, etc.) to interact with the device environment to fulfill human instructions.

Despite rapid progress, currently, most existing mobile LLM agents are confined to a completely passive paradigm: they only perform tasks upon receiving a clear instruction. This paradigm restricts their ability to proactively offer task suggestions and assistance in the absence of direct human instructions. If users have to formulate detailed instructions for every intent when interacting with mobile LLM agents, it will significantly increase the cognitive burden of mobile phone usage. Moreover, humans sometimes may not clearly express some latent needs. Therefore, mobile LLM agents need to be more proactive to provide users with more comprehensive and seamless services. Furthermore, the existing agents utilize almost exclusively user instructions as textual information

---

*Corresponding authors.

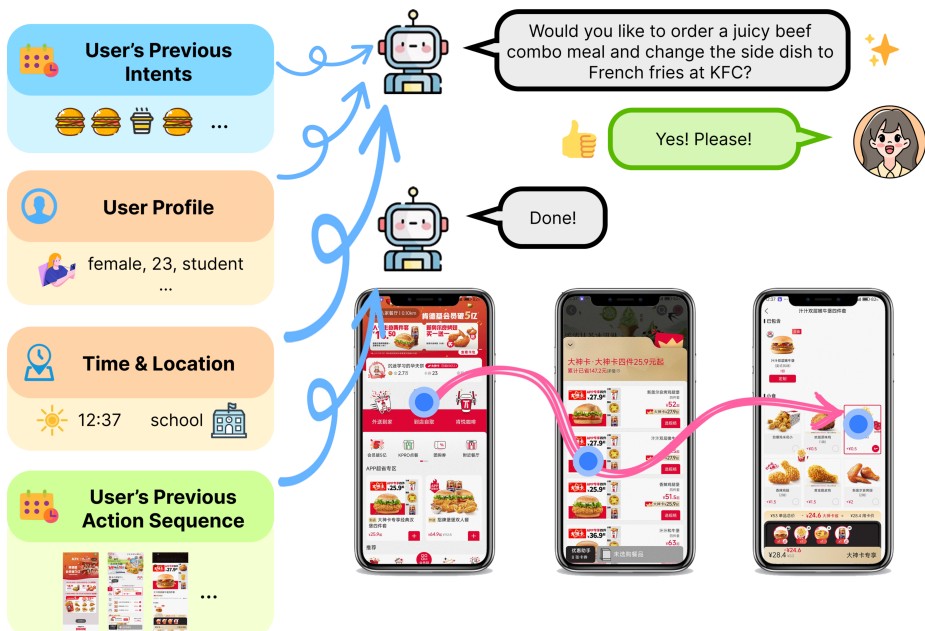

Figure 1: An overview task example in FingerTip 20K. The agent proactively offers task suggestions to the user and personalizes the execution of tasks in a way that aligns with the user's preferences.

when performing tasks, without taking into account any additional user-related information (e.g., time and location, user profile, user historical intents and actions), thus failing to provide personalized services to users. We argue that these limitations stem largely from the lack of suitable training data and standardized evaluation benchmarks that incorporate rich user-related information.

To comprehensively evaluate the proactive and personalized capabilities of mobile LLM agents, we propose the FingerTip 20K benchmark, which includes two new tracks: (i) proactive task suggestion, where the agent needs to integrate the user's past intents and the current environmental state to infer the user's potential current intent; (ii) personalized task execution, where the agent needs to refer to the user's past action preferences to execute current instructions. The overall task scenario we envision is shown in Figure 1. Since existing benchmarks do not provide user-related contextual information and historical data, we spent over one month collecting new diverse data from 95 users in their daily mobile phone usage, including 21,437 episodes covering 506 apps. We then conducted experiments on the FingerTip 20K benchmark to evaluate the capabilities of generalist models and GUI-control agents built on specifically designed models and found that there is still much room for improvement in their proactive and personalized capabilities. Current agents still find it hard to reach or surpass the human level. The best-performing model achieved a success rate of 12.8%, while humans reached 30.3%. We fine-tuned a small model using the collected data and achieved better results.

In summary, the main contributions of this work include:

- We propose the FingerTip 20K benchmark, which includes two brand-new tracks, to evaluate the ability of mobile LLM agents to proactively predict user intents and offer suggestions, as well as their ability to personalize task execution in accordance with user preferences.

- We collect large-scale user-oriented mobile GUI-control data, derived from scenarios in users' daily lives, which includes user-related contextual information and users' long-term mobile phone usage patterns.

- We evaluated the capabilities of generalist models and GUI-control-specific models on the FingerTip 20K benchmark, demonstrating the difficulty of the tracks we propose. The excellent performance of the model fine-tuned with our collected data highlights the potential of our approach in building more proactive and personalized mobile agents.

## 2 RELATED WORK

### 2.1 MOBILE GUI-CONTROL DATASETS AND BENCHMARKS

Table 1 compares FingerTip 20K to existing mobile GUI-control datasets and benchmarks (Chai et al., 2024; Li et al., 2024; Rawles et al., 2023; 2024; Xu et al., 2024a; Chai et al., 2025; Ran et al., 2025; Chen et al., 2024). These datasets typically represent each data instance through two core components: a textual task instruction and its corresponding operational demonstration. The demonstration is encoded as a sequence of interface interactions (e.g., clicking, typing, scrolling) accompanied by relevant screenshots. What differentiates them is mainly whether they are single-step (grounding instructions to UI elements on the screen), and whether they have supplemental View Hierarchy (VH) data for each screenshot. These datasets share some common drawbacks. Firstly, their task instructions are either pre-defined by authors or generated by LLMs, and it is questionable to what extent they can reflect the real intents of people using mobile phones in their daily lives. Additionally, they collect task demonstrations mainly by having annotators operate simulators on computers, which is not the real scenario of people using mobile phones. Finally, each data instance is isolated, lacking temporal correlation and contextual information related to the user.

Table 1: Comparison of FingerTip 20K to existing mobile GUI-control datasets and benchmarks.

| Dataset & Benchmark | #Episode | #Apps | #Avg steps | User-defined tasks? | Contextual info? | Historical data? | Task setting |
|---|---|---|---|---|---|---|---|
| Android Instruct | 10.5k | - | 9.0 | ✗ | ✗ | ✗ | execution |
| AMEX | 3046 | 192 | 12.8 | ✗ | ✗ | ✗ | execution |
| AndroidControl | 15283 | 833 | 5.5 | ✗ | ✗ | ✗ | execution |
| AitW | 715142 | 357 | 6.5 | ✗ | ✗ | ✗ | execution |
| AndroidWorld | 116 | 20 | - | ✗ | ✗ | ✗ | execution |
| AndroidLab | 138 | 9 | - | ✗ | ✗ | ✗ | execution |
| A3 | 201 | 20 | - | ✗ | ✗ | ✗ | execution |
| SPHINX | - | 100 | 8.1 | ✗ | ✗ | ✗ | execution |
| SPA-Bench | 340 | 58 | - | ✗ | ✗ | ✗ | execution |
| FingerTip 20K | 21437 | 506 | 11.1 | ✓ | ✓ | ✓ | proactive task suggestion & personalized task execution |

For benchmarks, the success rate is the most commonly used metric, and some studies also consider efficiency and cost. A common approach to assessing the success of a task is to determine whether essential states have been reached (Rawles et al., 2024; Zhang et al., 2024; Lee et al., 2024). Some studies also compare agents' actions to golden actions (Xing et al., 2024). However, these golden actions do not take into account potentially vast differences in user preferences, that is, the action sequences of different users to complete similar tasks may be very different. In addition, current benchmarks have similar task forms, that is, given an existing instruction, how to perform actions to complete it. To the best of our knowledge, there is no mobile LLM agent benchmark that discusses how to proactively suggest tasks based on user-related information when instructions are unknown.

### 2.2 MOBILE GUI-CONTROL AGENTS

Mobile GUI agents are designed to understand the UI and automate tasks on mobile apps in a manner similar to that of humans. Current agents leverage the extensive world knowledge and powerful embodied capabilities of multimodal large language models (MLLMs) for complex task planning and reasoning in multi-step GUI-control tasks. One notable approach is to directly guide generalist models like GPT-4v to perform tasks through extensive prompt engineering (Yan et al., 2023; Rawles et al., 2023; He et al., 2024; Koh et al., 2024; Kim et al., 2023; Zheng et al., 2023; Zhang et al., 2025; Wen et al., 2024). However, these methods require meticulously designed prompts to achieve the best results. Another research direction focuses on fine-tuning smaller models (Nakano et al., 2022; Qin et al., 2025; Hong et al., 2024; Xu et al., 2024b; Gur et al., 2023) on GUI-specific datasets to endow them with GUI grounding capabilities and the ability to break down high-level instructions, thereby enhancing their operational efficiency. Despite these advancements, most current agents are still confined to passively following explicit instructions and are unable to proactively predict user needs. Moreover, they do not take into account any user preferences when performing tasks. Some studies focus on proactively clarifying users' ambiguous instructions (Wu et al., 2021; Chen et al., 2020; Qian et al., 2024); however, these studies still require users to provide initial instructions. Proactive

Agent (Lu et al., 2024) predicts potential tasks by monitoring user activities and environmental states, but the input is text-only, and the task scenarios are mainly limited to computer or web environments rather than mobile ones.

# 3 PROBLEM FORMULATION

## 3.1 PROACTIVE TASK SUGGESTION

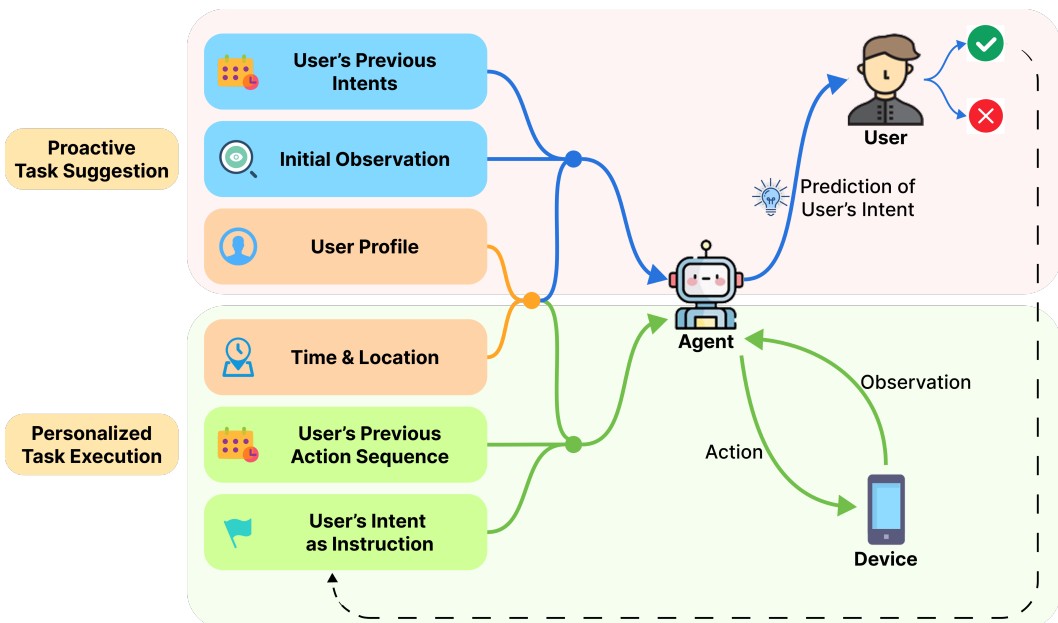

Figure 2: Demonstration of proactive task suggestion and personalized task execution.

In the FingerTip 20K benchmark we propose, unlike the evaluation tasks of traditional mobile LLM agent benchmarks that rely entirely on explicit instructions, we introduce a new task where the agent proactively predicts the user's current intent and proposes tasks suggestion that the user might want to perform, as shown in Figure 2. The agent's task is to generate an intent prediction $I$ based on the user profile $U$, the current time $T$, the current scenario $S$, the user's historical intents $I_{\text{history}}$, and the partial screenshots $O$ observed at present. This can be formalized as:

$$I = f\left(U,\ T,\ S,\ I_{\text{history}},\ O\right) \tag{1}$$

where $f$ represents the agent. $I$ is a sentence that unambiguously predicts the intent of the user. It should clearly state the name of the app that the user wants to use, and the final effect that the user wants to achieve. $U$ includes common user attributes such as age, sex, occupation, etc. $T$ represents the current timestamp, accurate to the second. $S$ represents the current scenario, expressed in common location categories. $I_{\text{history}}$ contains the user's historical intents in the recent period, up to 20 items, which may include the potential patterns and preferences of the user's mobile phone usage. $O$ includes the first few screenshots of the user's current actions (e.g., opening the home page of a certain app). We hope that the agent can utilize the above-mentioned user-related contextual information and historical intents to infer the user's potential intents, and thereby proactively offer helpful task suggestions.

## 3.2 PERSONALIZED TASK EXECUTION

In addition to proactive task suggestion, we also aim to evaluate the agent's ability to carry out tasks under the condition of explicit instructions, that is, when the user's intent is known. The setting of this part is similar to the existing benchmarks. The difference lies in that we additionally assess the agent's ability to execute tasks in a personalized manner specifically catering to the action preferences

of different users. Given user profile $U$, user intent $I_{\text{true}}$, user's historical actions $A_{\text{history}}$, agent's action sequence $A_{\text{agent}}$, and the current screenshot $O_t$ and the corresponding accessibility tree $AT_t$, the agent needs to perform the next action $A_{t+1}$, and then observe $O_{t+1}$ and $AT_{t+1}$. This can be formalized as:

$$A_{t+1},\ O_{t+1},\ AT_{t+1} = f\ (U,\ I_{\text{true}},\ A_{\text{history}},\ A_{\text{agent}},\ O_t,\ AT_t) \tag{2}$$

where $f$ represents the agent. $I_{\text{true}}$ is equivalent to the user's true intent that needs to be predicted in proactive task suggestion, and here it serves as the instruction to be executed. $A_{\text{history}}$ is the complete action sequence of the user when performing a similar task in the past, provided to the agent for in-context learning to imitate the user's action preferences. $A_{\text{agent}}$, on the other hand, is the action sequence $\{A_1, ..., A_t\}$ that the agent has already executed in the current task, helping the agent determine the progress of the task. The agent needs to constantly interact with the mobile phone environment until it believes that $I_{\text{true}}$ has been fulfilled. We hope that the final sequence of agent actions $A_{\text{agent}}$ can reflect the user's action preferences.

## 4 THE FINGERTIP 20K BENCHMARK

### 4.1 OVERVIEW

The motivation for FingerTip 20K data collection is to evaluate the dual tracks we have proposed, namely proactive task suggestion and personalized task execution. To this end, the most distinctive feature of the data should be user-oriented, containing sufficient user-related contextual information and being able to reflect the patterns and preferences of users in terms of intents and actions.

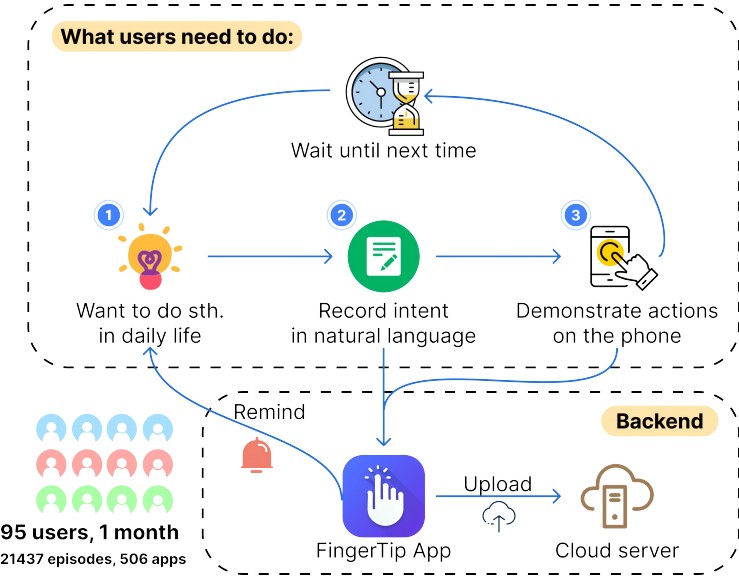

Figure 3: Data collection pipeline. Users record their intents and demonstrate actions by using the FingerTip APP in their daily mobile phone usage.

### 4.2 DATA COLLECTION

The data collection pipeline is shown in Figure 3. We first recruited 95 data collectors (hereinafter referred to as users) using Android phones through crowdsourcing, covering a wide range of device types and Android versions. Users were required to download an APP developed by us on their own daily used phones and use it to collect data. Specifically, whenever users had a real intent to use their phones in their daily lives, they could open the FingerTip APP, record their intent at that moment in one sentence, and select the location category they were in. Then, users needed to switch to the app involved in the intent they recorded and demonstrate the specific action sequence to complete this intent.

The FingerTip APP will automatically upload the intent they fill in (including time and location) and the demonstration process they provide (including screenshot sequences, corresponding accessibility tree XML file sequences, and UI action sequences) to the cloud server. This is regarded as the user collecting one piece of data. The APP may remind the user to collect data when they wake up the phone screen to prevent them from forgetting. Each user is required to use their phone to collect data for one month, with a maximum of 12 pieces of data uploaded per day. In this way, users can fully customize the data they upload. See Appendix A.3 and A.4 for more details on data collection and data format.

FingerTip APP is developed based on the accessibility features of the Android system. It can automatically record the type and coordinates, as well as optional text descriptions of each user action. The actions we collect are unified into an action space, as shown in Table 2. Among them, $finish$ is uniformly added to the last screenshot of all episodes.

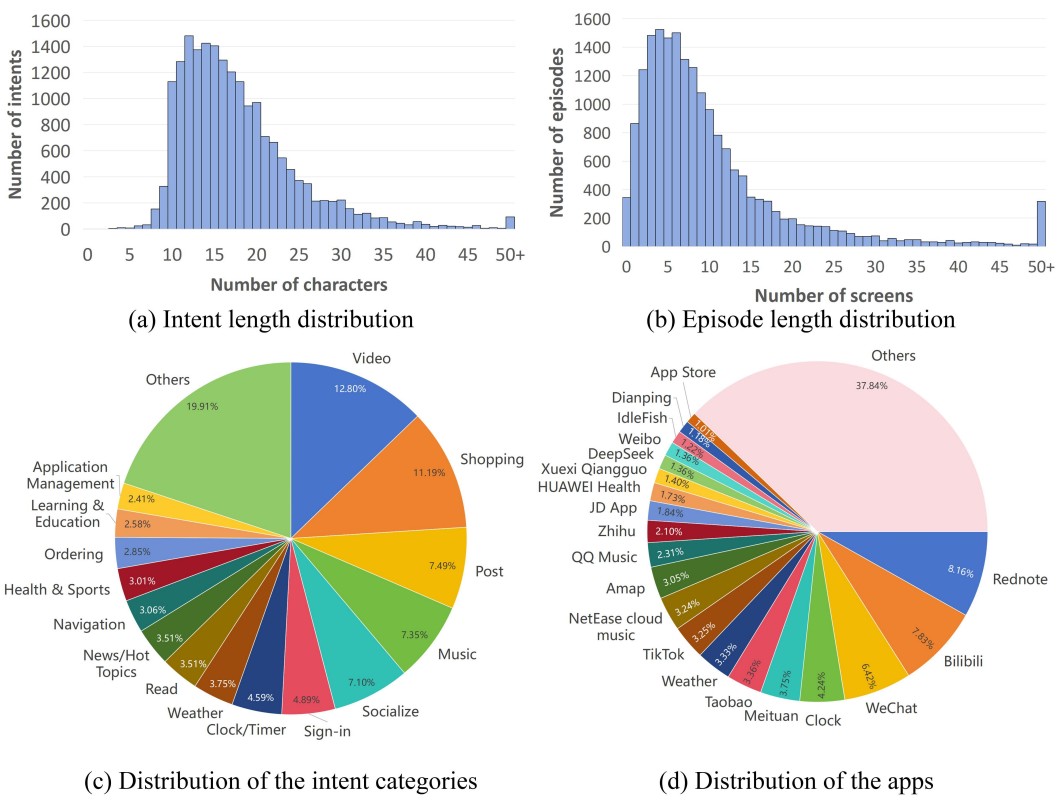

(a) Intent length distribution

(b) Episode length distribution

(c) Distribution of the intent categories

(d) Distribution of the apps

Figure 4: Dataset statistics and distribution. (a) The length distribution of the natural language intents recorded by users. (b) The distribution of the number of screenshots contained in each episode (i.e., the distribution of the number of action steps of users). (c) The distribution of all categories to which the intents belong. (d) The distribution of all apps involved in the data.

## 4.3 DATA STATISTICS

The summary of data statistics is presented in Table 1. Additionally, Figure 4 reports the distribution of user intent length, episode length, intent categories, and app name in all data. The intent categories are determined by DeepSeek-V3 (Liu et al., 2024).

## 4.4 PERSONALIZED ACTION ANALYSIS

To verify the personalized differences in actions among users of different types, we first simply classified users into different categories based on age groups. Then, we randomly sampled one piece of data from each of the 40 intent categories. For the action sequence of such a piece of data, we

Table 2: The action space of an agent when interacting with a mobile phone environment.

| Action | Parameter |
|---|---|
| click | coordinates=(x,y), content=" |
| long_click | coordinates=(x,y), content=" |
| type | text=" |
| scroll | coordinates=(x,y), direction=" |
| press_back | - |
| press_home | - |
| press_recent | - |
| wait | - |
| finish | - |

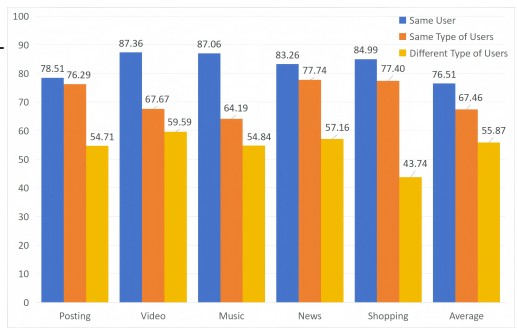

Figure 5: Personalized action analysis. We demonstrated the similarities of user action sequences in six intent categories. The similarities were higher among the same users or users of the same type, while the similarities between users of different types were lower.

calculated the Levenshtein similarity with the action sequence of the most intent-similar data from (i) the same user, (ii) users of the same type, and (iii) users of different type. All similarities were normalized to [0, 100] and plotted in Figure 5. It can be seen that even when performing similar intents, the similarity of action sequences with users of different type is significantly lower than that of the same user or users of the same type, indicating that user preferences on action sequences do exist and are measurable.

# 5 EXPERIMENTS

We conducted experiments on some generalist models and some GUI-control agents built on specifically designed models, evaluating their capabilities on the two tracks proposed in the FingerTip 20K benchmark and assessing their performance under different task difficulties. Additionally, we fine-tuned a model using the collected data. For details on the data splits (including the training set, validation set, and two test sets), please refer to Appendix A.5.

## 5.1 EXPERIMENTAL SETUP

**Proactive task suggestion** The LLMs we experiment with in this track include GPT-4.1, Qwen-VL-Max, DeepSeek-VL2 (Wu et al., 2024) and Qwen-2.5-VL-7B (Bai et al., 2025). We also introduce Qwen-2.5-VL-72B (Bai et al., 2025) to compare with the 7B version; and Qwen-QVQ-Max (thinking model) to compare with other non-thinking models. We set the temperature to zero for all models. For proactive task suggestion, the agent only needs one query to output the predicted intent. Since this is a brand new track proposed in our benchmark, there is no mature agent design available for direct use. We have designed a simple prompt to provide to all models evaluated in this track. This prompt contains all necessary inputs (see Section 3.1 and Appendix A.7.1).

**Personalized task execution** In this track, in addition to the generalist models mentioned above, we also experiment with three GUI-control agents built on specifically designed models, including Aguvis-7B (Xu et al., 2024b), CogAgent-9B (Hong et al., 2024) and UI-TARS-1.5-7B (Qin et al., 2025). We also introduce AutoDroid Wen et al. (2024) and AppAgent Zhang et al. (2025), two GUI-control agents based on prompt engineering (using GPT4.1 as the base model). For personalized task execution, the agent needs to interact with the environment in multiple steps to fulfill the user's instructions. We connect a physical phone to the computer via USB and use Android Debug Bridge (ADB) to provide this environment. Using an emulator would be a more convenient approach, but due to strict app control measures, most Chinese apps can only run on physical phones rather than emulators. For the generalist models, we designed a simple prompt to guide their output of the next action, with the action space consistent with Table 2. This prompt contains all necessary inputs (see Section 3.2 and Appendix A.7.2). For the GUI-control agents, they have specific format requirements

for input and output. To ensure normal output effects, their original prompts were used, and the input information in Section 3.2 was uniformly integrated into these original prompts. Their output was converted into a form consistent with the action space.

**Metrics** In proactive task suggestion track, the goal of the agent is to maximize the textual similarity between the output and the user's true intent. We use a pre-trained model, paraphrase-multilingual-MiniLM-L12-v2 (Reimers & Gurevych, 2019), to convert the agent's output and the user's true intent into embedding vectors and calculate their cosine similarity $S_1$. And, we calculate the Levenshtein similarity $S_2$ of these two strings. Both similarities are normalized to the range of [0, 1]. Finally, we take $Sim_1 = (S_1 + S_2)/2$ to comprehensively represent the text similarity. In addition to this numerical metric, we also use DeepSeek-V3 (Liu et al., 2024) to directly determine whether the agent's output and the user's true intent can be regarded as the same intent and provide a binary value to evaluate whether the agent successfully predicted the user's intent, thereby calculating the success rate $SR_1$.

In personalized task execution track, the primary goal of the agent is to execute user instructions in a personalized manner. We calculate the final success rate $SR_2$ by manually checking whether the environment state when the agent outputs $finished()$ matches the user's instructions. In addition, when the agent steps exceed 2.5 times the golden steps, the task is automatically considered a failure. Note that the path to successfully execute the user's instructions is not unique. The agent should also make the action sequence reflect the user's action preferences as much as possible. We do not require the agent's action at each step to be exactly the same as the user's golden action. Instead, we calculate the Levenshtein similarity $S_I$ of the agent's complete action sequence and the user's complete action sequence as two strings. Then, following the approach in Section 4.4, we take the data that is most similar to the current user's intent from the users of different type, and calculate the Levenshtein similarity $S_{II}$ of the agent's complete action sequence and this data's complete action sequence. Finally, we take the value $Sim_2 = S_I/S_{II}$. It is obvious that the larger this value is, the more similar the agent's action sequence is to that of the current user, and the more different it is from that of users of different type. In addition, we measure execution efficiency by comparing the agent steps with the user's golden steps to calculate the step ratio when the agent successfully execute the user's instructions. For the two tracks, we also tallied the average time and token count consumed per query to assess the model's cost.

## 5.2 OVERALL PERFORMANCE

The overall performance of the models we evaluated in proactive task suggestion is shown in Table 3. Note that here we set the number of $O$ (the first few screenshots of the user's current actions) to 0. This makes the task quite challenging. The thinking model Qwen-QVQ-Max surpassed GPT-4.1, achieving the best performance among the generalist models with $SR_1 = 12.8$ and $Sim_1 = 0.39$, but also took the longest time and the most tokens. From $SR_1$, it can be intuitively seen that the success rate of all models in predicting the user's intent is very low. Additionally, we conducted a user study where 20 human annotators (distinct from the users who collected the data) labeled a subset of the test set (a total of 400 episodes), achieving a success rate of 30.3%. This highlights the significant gap between the existing models and human in proactive task suggestion capabilities.

Table 3: Overall performance of proactive task suggestion.

| Model | $SR_1$ (%) | $Sim_1$ | Time (sec) | Token |
|---|---|---|---|---|
| GPT-4.1 | 7.2 | 0.35 | 5.64 | 796 |
| Qwen-VL-Max | 6.9 | 0.33 | 1.98 | 950 |
| Deepseek-VL2 | 4.3 | 0.25 | **0.71** | **743** |
| Qwen-2.5-VL-7B | 3.1 | 0.25 | 0.78 | 943 |
| Qwen-2.5-VL-72B | 7.0 | 0.31 | 5.45 | 963 |
| Qwen-QVQ-Max | **12.8** | **0.39** | 10.60 | 2335 |
| Human | 30.3 | 0.57 | - | - |

The overall performance of the models we evaluated in personalized task execution is shown in Table 4. Qwen-QVQ-Max and UI-TARS-1.5-7B achieved the best performance among the generalist models and GUI-control models respectively. AppAgent achieved the best performance among all

Table 4: Overall performance of personalized task execution.

| Model | $SR_2$ (%) | $Sim_2$ | Step Ratio | Time (sec) | Token |
|---|---|---|---|---|---|
| GPT-4.1 | 5.5 | 0.98 | 1.98 | 8.02 | 2912 |
| Qwen-VL-Max | 4.5 | **1.07** | 2.06 | 4.17 | 2304 |
| Deepseek-VL2 | 1.0 | 0.93 | 2.19 | **3.46** | 2130 |
| Qwen-2.5-VL-7B | 1.5 | 0.95 | 2.16 | 3.66 | 2213 |
| Qwen-2.5-VL-72B | 4.0 | 0.96 | 2.05 | 9.31 | **2018** |
| Qwen-QVQ-Max | **9.5** | 1.04 | **1.94** | 15.60 | 3048 |
| AutoDroid | 10.5 | 1.08 | 1.29 | 22.20 | **3123** |
| AppAgent | **11.0** | **1.12** | **1.13** | **19.74** | 3853 |
| Aguvis-7B | 20.5 | 1.02 | 1.38 | **6.86** | 2494 |
| CogAgent-9B | 18.0 | 0.92 | 1.73 | 12.54 | 2808 |
| UI-TARS-1.5-7B | **38.5** | **1.06** | **1.22** | 10.15 | **2440** |

models in $Sim_2$ and step ratio, possibly due to its proficiency in learning from human demonstrations, but time and token costs also increased significantly. The $SR_2$ of the generalist models were all very low, mainly due to their lack of precise GUI grounding ability, which led to incorrect UI coordinates being output even when they could correctly analyze the next action, thus failing to interact with the environment accurately. In contrast, the GUI-control models, having undergone targeted training, had stronger abilities to execute instructions and interact with the UI environment, resulting in higher $SR_2$, with UI-TARS-1.5-7B having the highest at 38.5. However, the $Sim_2$ of all models were approximately 1, indicating that the agent's action sequence did not favor either the current user or users of different type. This might suggest that the agent tends to complete tasks in a general way without catering to the specific action preferences of users, thus failing to complete tasks in a personalized manner.

## 5.3 EFFECT OF TASK DIFFICULTY

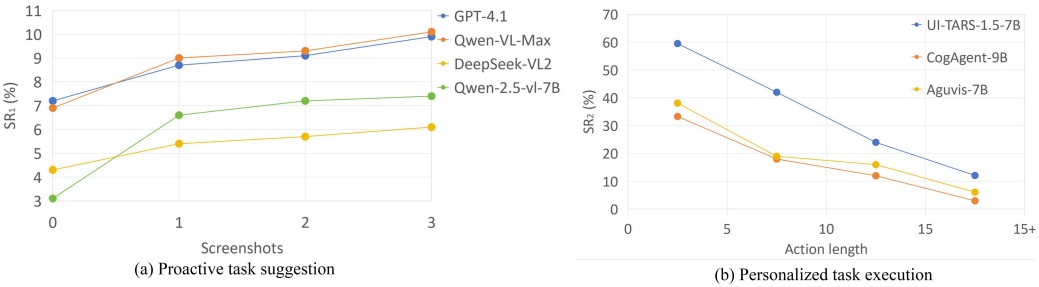

(a) Proactive task suggestion

(b) Personalized task execution

Figure 6: Performance under different task difficulties. (a) The variation of $SR_1$ under different numbers of input screenshots. (b) The variation of $SR_2$ under different action lengths.

We experiment with the models' performance under different task difficulty levels. For proactive task suggestion (see Figure 6.a), we varied the number of $O$ (the first few screenshots of the user's current actions). The $SR_1$ of all models increased as the number of screenshots increased. This was expected. Clearly, if the agent knew the first screenshot of the user's current action, it could basically infer which app the user was using, thereby significantly narrowing the range of the user's intent. With the second and third screenshots, the agent could further narrow the user's intent range by analyzing the actions therein (e.g. clicking the search box).

For personalized task execution (see Figure 6.b), we calculated the $SR_2$ of GUI-control models on different subsets of action length (i.e., the number of action steps) in the test set. It can be seen that as the action length increases, the $SR_2$ decreases. This is in line with expectations, as the greater the action length required to complete a certain instruction, the more complex the instruction is, and the more difficult it is to complete.

## 5.4 EFFECT OF FINE-TUNING

We fine-tuned Qwen-2.5-VL-7B and adopted the parameter-efficient fine-tuning method of LoRA, with the LoRA rank set to 4 or 64. Following the method of sampling the test set, we randomly sampled 1,000 data episodes from the training set for fine-tuning. These data covered all users, and the proportion of data for each user was the same as their proportion in the training set. We also used the complete training set (16,000 episodes) for fine-tuning. The data episodes were reorganized according to the input and output formats of the two tracks, respectively. The prompts used in fine-tuning are the same as those we designed for generalist models. Finally, we trained separately on two tracks and obtained two fine-tuned models, each suitable for one of the two tracks.

Table 5: Performance of fine-tuned models. In square brackets [X] we report the performance increase from the un-fine-tuned Qwen-2.5-VL-7B.

| Model | Proactive task suggestion | | Personalized task execution | | |
|---|---|---|---|---|---|
| | $SR_1(\%)$ | $Sim_1$ | $SR_2(\%)$ | $Sim_2$ | Step Ratio |
| Qwen-2.5-VL-7B | 3.1 | 0.25 | 1.5 | 0.95 | 2.16 |
| Qwen-2.5-VL-7B-FT-1k-r4 | 9.7 [+6.6] | 0.49 [+0.24] | 12.5 [+11.0] | 1.21 [+0.26] | 1.17 [-0.99] |
| Qwen-2.5-VL-7B-FT-1k-r64 | 11.8 [+8.7] | 0.50 [+0.25] | 12.5 [+11.0] | 1.26 [+0.31] | 1.17 [-0.99] |
| Qwen-2.5-VL-7B-FT-all-r4 | 20.3 [+17.2] | 0.52 [+0.27] | 15.0 [+13.5] | 1.32 [+0.37] | 1.15 [-1.01] |
| Qwen-2.5-VL-7B-FT-all-r64 | **26.0** [+22.9] | **0.55** [+0.30] | 15.5 [+14.0] | **1.42** [+0.47] | **1.13** [-1.03] |
| Qwen-QVQ-Max | 12.8 | 0.39 | 9.5 | 1.04 | 1.94 |
| UI-TARS-1.5-7B | - | - | **38.5** | 1.06 | 1.22 |

The performance of fine-tuned models on the two tracks is shown in Table 5. Despite using a smaller model and less training data, the fine-tuned models achieved significant performance improvements in all main metrics. Increasing the LoRA rank or the amount of training data both improve the model's performance, with the increase in training data having a particularly significant effect. In proactive task suggestion, compared with the best-performing generalist model Qwen-QVQ-Max, our fine-tuned models achieved better performance in both $SR_1$ and $Sim_1$. In personalized task execution, compared with the best-performing UI-TARS-1.5-7B, our fine-tuned models had a lower success rate $SR_2$. We consider this acceptable because UI-TARS is a model specifically designed and extensively trained for GUI grounding and GUI control, and thus has a more general instruction execution capability. However, our fine-tuned models had a significantly higher $Sim_2$, indicating that the action paths they select may not be optimal but are closer to the user's action preferences. When trained on the entire training set with a LoRA rank of 64, Qwen-2.5-VL-7B outperforms all the un-fine-tuned models in the experiment in terms of $SR_1$, $Sim_1$, and $Sim_2$, achieving the best performance. Overall, the models fine-tuned on our collected data demonstrated stronger proactivity and personalization capabilities, being able to utilize user-related contextual information to extract potential intent patterns and action preferences from the user's past intents and actions, which existing models have not or find it difficult to consider.

**More experiments** In Appendix A.6 we conducted more experiments to study the influence of other factors.

## 6 CONCLUSIONS

We present FingerTip 20K, a benchmark advancing mobile LLM agents toward proactive task suggestion and personalized task execution. Our data captures longitudinal user interactions, enriched with contextual information to model user-specific patterns. Experiments reveal significant gaps in existing models' ability to mine such patterns. Fine-tuning Qwen-2.5-VL-7B on our data improved suggestion success rate while better aligning actions with user preferences, demonstrating the value of user-oriented training. This work establishes critical infrastructure for developing mobile agents that anticipate user needs and adapt to user action preferences.

## 7 ETHICS STATEMENT

Our data collection involves human participants. We detail our data collection process and the multiple measures we have taken to reduce the risk of privacy leakage in Appendix A.3. We also discuss the broader impacts of this study in Appendix A.2.

## 8 REPRODUCIBILITY STATEMENT

To ensure the reproducibility of our work, we provide all the necessary resources and code used in this paper. All adopted models are fully open source or publicly accessible. Our project code, including the data format, data splits, and evaluation process of FingerTip 20K, can be publicly accessed via the following anonymous link: `https://github.com/tsinghua-fib-lab/FingerTip-20K`.

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

# A  APPENDIX

## A.1  LIMITATIONS

Our study has several limitations. Firstly, all 95 contributors live in mainland China, and mainly interact with Chinese third-party apps. The recorded linguistic habits, UI layouts and action patterns may differ markedly from other regions. Secondly, our LoRA fine-tuning uses only a single 7B model. Due to cost constraints, we did not conduct larger-scale fine-tuning experiments. Finally, we assume that screenshots can be stored and shared after anonymization. In practice, fine-grained UI traces can still contain unique visual features that allow re-identification. Techniques such as selective redaction or synthetic replay should be explored before large-scale deployment.

## A.2  BROADER IMPACTS

FingerTip 20K aims to advance mobile agents that anticipate user needs and adapt to individual preferences. If developed responsibly, such agents could reduce the interaction barrier for elderly or motor-impaired users, reduce screen time by automating repetitive tasks, and serve as a test bed for privacy-preserving personalization research. At the same time, the technology entails risks. Continuous screen capture combined with explicit user profiles gives models an intimate view of personal life. An attacker compromising the agent, or a service provider lacking strong governance, could reconstruct sensitive behaviors, contacts or locations. We encourage future work on on-device processing, differential privacy and audit mechanisms.

## A.3  DATA COLLECTION

The data collection was carried out through crowdsourcing, and participants were paid in accordance with the living wage laws of their country. Participants consist of one-third undergraduates, one-third postgraduates, and one-third employed individuals, including 54 males and 41 females, whose ages range from 18 to 60, with an average age of 25.9. Participants filled out a questionnaire, which collected their user profiles. Participants were informed of the expected use of the collected data and signed a data usage agreement. They were asked not to upload any data related to private information. We provided participants with detailed guidance documents and video tutorials on how to operate the FingerTip APP for data collection. All participants went through a training phase during which they became familiar with the FingerTip APP. They were encouraged to avoid using overly simplified or ambiguous language to collect clear and useful intent descriptions. They were clearly informed that they should not perform redundant or useless operations during the demonstration process, and the operation speed should not be too fast to avoid frequent repetitive operations. However, minor noisy operations (e.g., users making a typo or accidentally touching advertisements) are realistic situations in human interaction. A robust agent must be able to handle such scenarios. Even if the demonstrations are not collected from daily life but by recruiting annotators to perform operations in a simulator like existing datasets, such noise cannot be completely avoided. Therefore, we allow for its existence. During data collection, we conducted multiple timed quality checks on the data submitted by each participant and manually deleted the low-quality data. We also provided quality feedback to the corresponding participants, reminding them how to submit higher-quality data.

It should be noted that the FingerTip APP only collects data when participants actively use it. It does not automatically collect data at other times. Participants can check or delete the data they upload at any time. We conducted two rounds of inspections. We first manually inspected the data and removed those that obviously involved privacy. Then, we used Qwen-VL-Max to examine the first and last screenshots of each episode and determine whether it involved privacy. Those episodes marked as potentially involving privacy were then rechecked by humans.

Our primary goal for collecting the data was to capture deep and longitudinal user interactions in daily life settings. We believe that this context-rich dataset, even from a single region, provides a crucial foundation for the novel tasks of proactive task suggestion and personalized task execution. Considering the cost, we did not collect data in other regions. To our knowledge, previous datasets such as Rawles et al. (2023); Li et al. (2024); Chen et al. (2024) also contain a single language and UI ecosystem. We believe that this is a sufficient start for a first-of-its-kind study. However, user diversity is a crucial aspect in ensuring the global generalizability of our findings. To facilitate

broader research, we plan to open source our APP for data collection. It can run on any (new version) Android personal phone, providing support for data collection in other regions and languages. We believe that our data collection methods and evaluation methods are universal.

## A.4 DATA FORMAT

Our data is released at `https://github.com/tsinghua-fib-lab/FingerTip-20K`. The data contains several folders named with numbers (i.e. user IDs), and each of these folders contains multiple folders named with timestamps (e.g., 20250309_133115), representing all the data episodes submitted by that user. For each data episode, the following information is included:

- *screenshots*: a list of screenshots for each observation encoded as JPGs.
- *accessibility trees*: a list of Android accessibility tree XML files for each observation.
- *actions*: a list of actions represented in the form of JSON dictionaries. Each screenshot corresponds to an action.
- *intent_description*: the user's true intent in this episode.
- *user_id*: the unique integer identifier of the user to whom this episode belongs. This information can be used to retrieve the corresponding user's user profile.
- *time*: the timestamp when this episode was collected.
- *scenario*: the category of location where the user was when this episode was collected.
- *app*: the name of the activity running when the episode was collected. This information is only used to launch the corresponding app in personalized task execution and is not provided to the LLM agent.

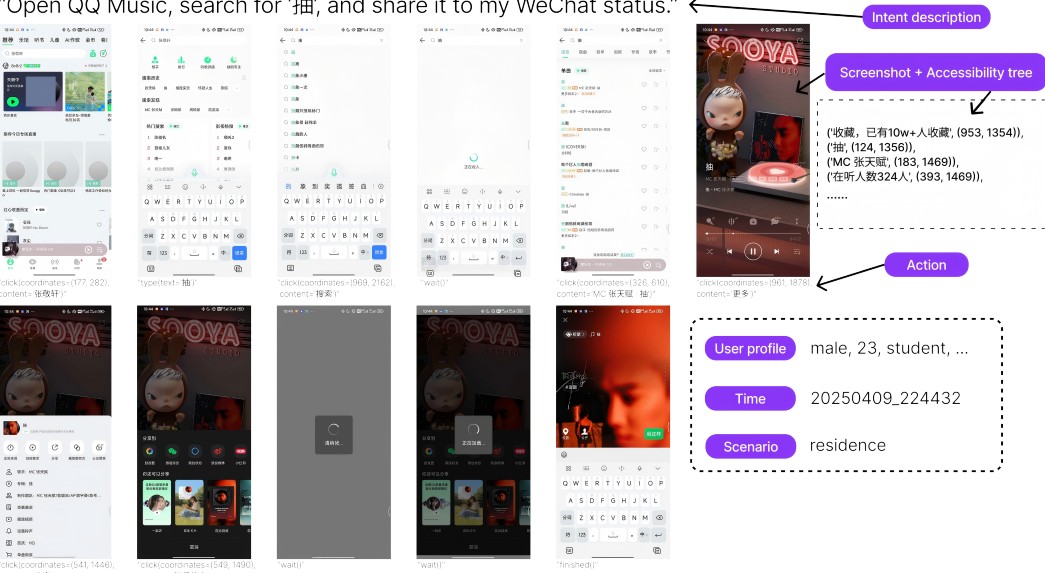

Figure 7: An example data episode from FingerTip 20K.

The example of an episode from FingerTip 20K is shown in Figure 7.

**Accessibility trees**   Note that when using accessibility trees, the LLM agent utilizes a list of all accessible UI elements and their coordinates corresponding to a certain screenshot, which is extracted from the metadata XML file through a Python function.

**User profile**   The types of information included in user profiles and an example can be seen in Table 6.

Table 6: User profile example.

| Field | user_id | sex | age | occupation | address | marital_status | phone_brand |
|-------|---------|-----|-----|------------|---------|----------------|-------------|
| Example | 55 | male | 20 | student | Beijing | single | Huawei |

**Scenario** When users record their intents with the FingerTip APP, they need to select the category of the location they are in. Specifically, they can choose from the following 12 common categories: residence, office, school, dining place, shopping mall, medical institution, entertainment and leisure venue, sports venue, cultural venue, transportation, urban street, and natural outdoor spaces. If users think that none of these categories can describe the location they are in, they can fill in a new category on their own.

## A.5 DATA SPLITS

Table 7: Details on FingerTip 20K train, validation and test splits. For each split, we report the number of episodes, the number of screenshots, the number of apps, and the number of intent categories it contains.

| Split | # Episodes | # Screens | # Apps | # Categories |
|-------|-----------|-----------|--------|--------------|
| Train | 16000 | 177674 | 460 | 40 |
| Vali | 4411 | 32859 | 41 | 27 |
| Test-suggestion | 1000 | 10412 | 155 | 38 |
| Test-execution | 200 | 2074 | 68 | 31 |

We created a training set, a validation set, and two test sets. The number of episodes and features in these sets are detailed in Table 7. Please note that the two test sets contain partially overlapping episodes. The test sets were formed by randomly sampling the last 20% of the data sorted by time of each user, and then concatenated to ensure coverage of all users and that the proportion of data from each user in the test sets is equal to their proportion in all data. These test sets were used in all main experiments. The collection method of the training set is similar to that of the test sets, except that it is sampled from the first 60% of the data.

## A.6 SUPPLEMENTARY EXPERIMENT RESULTS

### A.6.1 OUT-OF-DOMAIN GENERALIZATION

To explore generalizability, we randomly sampled from the original test set and obtained three small test subsets, which are: (1) User-unseen, containing 126 episodes from 3 users. All data of these 3 users in the training set were removed. (2) App-unseen, containing 106 episodes from 7 apps. All data of these 7 apps in the training set were removed. (3) Intent-unseen, containing 99 episodes from 4 intent categories. All data of these 4 intent categories in the training set were removed. The filtered training set has 14,706 episodes, and these data were used to re-fine-tune Qwen-2.5-VL-7B, with the LoRA rank set to 4. The fine-tuned model was tested on these three out-of-domain test sets and the original test set, and the results are shown in Table 8.

Table 8: Performance of the fine-tuned model on out-of-domain test sets.

| Test set | Proactive task suggestion | | Personalized task execution | | |
|----------|---------------------------|----------|-----------------------------|----------|------------|
| | $SR_1(\%)$ | $Sim_1$ | $SR_2(\%)$ | $Sim_2$ | Step Ratio |
| Original test set | 19.9 | 0.51 | 13.5 | 1.29 | 1.18 |
| User-unseen | 15.1 | 0.50 | 13.2 | 1.29 | 1.21 |
| App-unseen | 14.2 | 0.49 | 12.7 | 1.22 | 1.23 |
| Intent-unseen | 15.2 | 0.51 | 13.1 | 1.27 | 1.21 |

When tested on new users, new apps, and new intent categories that have not been seen in the training set, the decline in model performance is not particularly severe. This indicates that the model

fine-tuned on partial data has certain generalization ability and robustness, and can maintain good proactive task suggestion and personalized task execution capabilities in unseen data as well.

### A.6.2 CONNECTION BETWEEN TWO TRACKS

We believe that proactive task suggestion and personalized task execution are both crucial capabilities for an agent to act as a user-oriented intelligent assistant. In practical applications, it first predicts the user's needs and then fulfills them in a way preferred by the user, thereby facilitating the user's more convenient use of the mobile phone and demonstrating a kind of collaborative connection. However, the two tracks are conceptually distinct and emphasize different capabilities. Proactive task suggestion places more emphasis on the agent's ability to predict the user's intents in advance, rather than passively responding to the user's clear instructions, that is, understanding "what the user wants to do". Personalized task execution places more emphasis on aligning the agent's behavior with the user's preferences during the known instruction execution process, rather than standardizing the task execution, that is, understanding "how the user does it". In the fine-tuning of Section 5.4, we trained separately on two tracks and obtained two fine-tuned models, each suitable for one of the two tracks. Now we test these two models on the opposite track from the training data. Additionally, we jointly fine-tuned a model (trained on both tracks), and the results are shown in Table 9.

Table 9: Performance of the separately fine-tuned model and the jointly fine-tuned model.

| Model | Proactive task suggestion | | Personalized task execution | | |
|---|---|---|---|---|---|
| | $SR_1(\%)$ | $Sim_1$ | $SR_2(\%)$ | $Sim_2$ | Step Ratio |
| Qwen-2.5-VL-7B | 3.1 | 0.25 | 1.5 | 0.95 | 2.16 |
| Qwen-2.5-VL-7B-FT | **9.7** | **0.49** | **12.5** | **1.21** | **1.17** |
| Qwen-2.5-VL-7B-FT-proactive | **9.7** | **0.49** | 1.0 | 0.97 | 2.20 |
| Qwen-2.5-VL-7B-FT-personalized | 2.9 | 0.25 | **12.5** | **1.21** | **1.17** |
| Qwen-2.5-VL-7B-FT-joint | 9.2 | 0.46 | 11.0 | 1.18 | 1.18 |

The model fine-tuned on one track did not bring about performance improvement when tested on the other track; instead, there was a performance decline. The performance of the jointly fine-tuned model also slightly declined compared to the separately fine-tuned models. This indicates that the two tracks test two different abilities, and it is necessary to train and evaluate them separately.

### A.6.3 CONTRIBUTION OF SCREENSHOTS AND HISTORICAL INTENTS

Our intention for the main results in Table 3 was to establish a baseline for the most challenging version of proactive task suggestion, where the agent has zero screenshots and must rely solely on historical and contextual data. This highlights the inherent difficulty of the task. To explore the performance of the agent under more screenshots or more historical information, we supplemented the experiments and obtained the following data in Table 10 (all using GPT4.1). Besides, we have already demonstrated the variation of performance with the number of screenshots in Figure 6.a.

Table 10: Performance of proactive task suggestion under different number of input screenshots or historical intents.

| Setting | $SR_1$ (%) | $Sim_1$ |
|---|---|---|
| 0 screenshot + 20 $I_{\text{history}}$ | 7.2 | 0.35 |
| 0 screenshot + All $I_{\text{history}}$ | 9.6 | 0.38 |
| 3 screenshots + No $I_{\text{history}}$ | 4.3 | 0.45 |
| 3 screenshots + 20 $I_{\text{history}}$ | 9.9 | 0.53 |
| 3 screenshots + All $I_{\text{history}}$ | **13.8** | **0.55** |

0 screenshot + 20 $I_{\text{history}}$ are the results we present in Table 3. For All $I_{\text{history}}$, we use DeepSeek-V3 to summarize the 20 most relevant historical intents of the user to the current time and scenario among all historical intents. Additionally, we also tested the results of providing 3 initial screenshots and mixing them with All $I_{\text{history}}$. Both providing more screenshots and historical information can improve performance, but there is still much room for improvement. Offering more screenshots would lose the predictive meaning of this task and significantly increase costs. We hope that the agent

can complete proactive task suggestion by relying on as few screenshots and historical information as possible. When $I_{history}$ was removed (cold-start users) while keeping three screenshots visible, the success rate dropped to 4.3%, indicating a significant performance decline. It is evident that historical intents are crucial for predicting current intents, and relying solely on screenshots cannot effectively accomplish proactive task suggestion.

### A.6.4    CONTRIBUTION OF CONTEXTUAL INFORMATION

To study the contribution of each contextual information in the input to the proactive task suggestion, we supplemented the ablation study (all using GPT4.1) and obtained the following results in Table 11.

Table 11: Performance of proactive task suggestion under different contextual information.

| Setting | $SR_1$ (%) | $Sim_1$ |
|---|---|---|
| w/ User profile, Time, Scenario | **7.2** | **0.35** |
| w/o User profile | 6.5 | 0.32 |
| w/o Scenario | 6.1 | 0.31 |
| w/o Time | 4.1 | 0.28 |

w/ User profile, Time, Scenario are the results we present in Table 3. Eliminating User profile, Scenario, and Time all lead to performance degradation, among which the elimination of Time causes the most significant decline, indicating that time might be the most crucial factor in the patterns of user intent.

### A.6.5    EFFECT OF THE PROBABILITY SETTING

Our data is longitudinal and collected over one month. This means that we often capture multiple instances of similar intents from the same user. This structure is precisely what allows for modeling user preferences and "habitual" intents. That is to say, within a specific time period of a day, a specific user's intents roughly follow a fixed probability distribution. We first separate all the intents of the same user by time periods (e.g., dividing a day into 24 time periods by hour). Then, we convert all the intents within the same time period into embedding vectors using paraphrase-multilingual-MiniLM-L12-v2Reimers & Gurevych (2019) and cluster them based on distance. All semantically similar intents are regarded as one category. If the number of intents in a certain category is larger, it indicates that the probability of the user generating this type of intent during this time period is higher. In this way, we obtain the probability distribution of intents (e.g., the user has a 35% probability of ordering a hamburger for delivery and a 22% probability of playing music from a self-built playlist between 12:00 and 13:00...). For each user, a unique probability distribution of intents can be calculated through the above method.

We re-executed the proactive task suggestion experiment by having GPT4.1 output the probability distribution of the user's intents instead of a single intent. The calculation method of $SR_1$ was changed to be successful as long as one of the top three intents in the output probability distribution could be regarded as the same as the user's true intent. The calculation method of $Sim_1$ was changed to the cosine similarity between the output probability distribution's embedding vector and the true probability distribution's embedding vector. It can be seen in Table 12 that by outputting the probability distribution, the agent provides multiple possible task suggestions, which is more likely to succeed than only outputting a single task suggestion.

Table 12: Performance of proactive task suggestion under a probability setting.

| Setting | $SR_1$ (%) | $Sim_1$ |
|---|---|---|
| Output a single intent directly | 7.2 | 0.35 |
| Output the probability distribution of intents | 11.1 | 0.42 |

### A.6.6    VALIDITY OF $Sim_2$

$Sim_2$ is an automated metric for personalization. To quantitatively analyze the correlation between $Sim_2$ and users' subjective experience, we conducted a user study. Specifically, for four models

in Table 5, we combined their output (i.e., the complete action sequences output by the models) on the personalized task execution test set (200 episodes) with the users' true action sequences. Each episode has one ground truth action sequence and four randomly ordered action sequences output by the models. Then, we asked the users corresponding to these 200 episodes to rate the four models' action sequences on a five-point scale. The rating principle was whether the action sequence was personalized to execute the task according to the user's unique habits and preferences, even if it might not have been ultimately successful. Then, we calculated the average rating of the four models and compared it with their $Sim_2$. The results are shown in Table 13.

Table 13: Comparison of $Sim_2$ and user ratings in personalized task execution.

| Model | $Sim_2$ | User Rating |
|---|---|---|
| Qwen-2.5-VL-7B | 0.95 | 2.42 |
| Qwen-2.5-VL-7B-FT | **1.21** | **3.35** |
| GPT-4.1 | 0.98 | 2.55 |
| UI-TARS-1.5-7B | 1.06 | 2.72 |

The user rating increases with the increase of $Sim_2$, indicating a certain positive correlation between $Sim_2$ and users' subjective personalized experience. The fine-tuned model has the highest $Sim_2$ and its user rating also reached the highest 3.35 points, indicating that fine-tuning on our data indeed improved the model's personalization ability.

### A.6.7 EFFECT OF SIMILAR OR SAME ACTION SEQUENCE

In personalized task execution, we provide the agent with an action sequence of a similar task for in-context learning. However, this similar task might be the same as the current one, as the user has performed the same task before, and the agent might cheat on the same task. We used DeepSeek-V3 to determine whether the retrieved similar tasks and the current task could be regarded as the same task. If they were the same, we moved on to the next similar task until they could no longer be considered the same. Using this method, we re-conducted the experiment on UI-TARS-1.5-7B, and the performance in Table 14 showed no significant difference from the original. Therefore, we believe there is no obvious cheating phenomenon. While tasks may be the same, the exact UI states are unlikely to be identical, so is the action sequence. The goal is for the agent to generalize a user's style of interaction, not replicate a specific trace.

Table 14: Performance of personalized task execution under different historical action sequences.

| Setting | $SR_2$ (%) | $Sim_2$ | Step Ratio |
|---|---|---|---|
| most similar task | 38.5 | 1.06 | 1.22 |
| most similar (not the same) task | 37.5 | 1.03 | 1.23 |

## A.7 PROMPTS FOR THE LLM AGENTS

### A.7.1 PROMPT FOR PROACTIVE TASK SUGGESTION

```
You are an Android GUI agent. You are given the first few screenshots of
the user's action (arranged in chronological order) and some
supplementary information. You need to infer the user's intent.

## Input
User_profile: {profile}
Time: {time}
Scenario: {scenario}
Previous_intents: {previous_intents}

## Note
- Express the user's intent unambiguously in one Chinese sentence,
including all necessary information.
- Clearly state the name of the app which the user is using, and the
final effect the user wants to achieve.
- Previous_intents contains the user's intents at certain times and in
certain scenarios in the past.
- Do not output anything other than the user's intent.

The user's intent:
```

### A.7.2 PROMPT FOR PERSONALIZED TASK EXECUTION

```
You are an Android GUI agent. You are given an instruction and current
screenshot and some supplementary information. You need to perform the
next action to complete the instruction.

## Input
User_instruction: {instruction}
User_profile: {profile}
Screen_width_height: {size}
Screen_description: {screen_description}
Actions_reference: {actions_reference}
Previous_actions: {previous_actions}

## Action Space
click(coordinates=(x,y), content='')
long_click(coordinates=(x,y), content='')
type(text='')
scroll(coordinates=(x,y), direction='down or up or right or left')
press_back()
press_home()
press_recent()
wait()
finished()

## Note
- 'coordinates' should represent the coordinates of the click point. The
origin is the upper left corner of the screenshot, with x increasing to
the right and y increasing downward.
- 'content' should represent the original text at the click point or the
description of the icon, usually in Chinese.
- 'text' should represent all the original text that the user intends to
input. (usually in Chinese, and usually included in User_instruction)
- 'press_back()', 'press_home()', 'press_recent()' means that go to
previous screen, home screen, recent apps screen, respectively.
- 'wait()' means that wait until the next observation is received. This
usually occurs during loading or switching windows.
- 'finished()' means that the instruction is completed.
```

```
- Screen_description contains some correct 'content' and 'coordinates' of
 the UI, which can be directly referenced.
- Actions_reference represents the complete sequence of actions that the
user performed when executing a similar instruction in the past, which
can be used for reference.
- Previous_actions contains the sequence of actions you have already
performed under the current instruction.
- Only one action in Action Space can be taken. Do not output anything
other than the action to take.

The action to take:
```

