# OpenReview forum: "FingerTip 20K: A Benchmark for Proactive and Personalized Mobile LLM Agents"
_ICLR.cc/2026/Conference — ICLR 2026 Poster_

### Official Review · Reviewer_NkBD · 2025-10-30

**Soundness:** 3
**Presentation:** 2
**Contribution:** 3
**Rating:** 4
**Confidence:** 4

**Summary:**

This paper introduces FingerTip 20K, a novel benchmark designed to evaluate the proactive and personalized capabilities of mobile GUI-control LLM agents. The contributions of the paper are clear — it presents 21,437 episodes collected from 95 real users using their own Android phones in daily life, covering 506 apps. This dataset is highly valuable for research on user-centric GUI agents.

**Strengths:**

1. Real-World, Longitudinal Data. The dataset is collected from real users on their own Android phones (rather than simulators), which represents the most prominent contribution of this work.

2. Novel Benchmark. The paper introduces two clearly defined new tracks — proactive task suggestion and personalized task execution, both of which address important challenges in GUI-based applications.

3. Experimental Evaluation. The benchmark evaluates a wide range of models, including GPT-4.1, Qwen-VL, DeepSeek, CogAgent, UI-TARS, and others.

**Weaknesses:**

1. All data contributors are from mainland China and use Chinese apps, which raises concerns about cross-cultural generalization and UI diversity.

2. Only 1,000 episodes were used to fine-tune a single 7B model with LoRA (rank 4), making the experimental validation somewhat narrow.

3. The model’s performance across both tracks remains generally poor, and fine-tuning does not effectively address this issue. Moreover, the paper lacks further analysis to explain these results.

**Questions:**

1. It is unclear why the personalized track is evaluated only in the online setting. If users employ different mobile systems, system-level inconsistencies may arise during online testing. More importantly, the online evaluation currently lacks an objective and standardized assessment mechanism, making reproducibility and fairness difficult to ensure.
2. The fine-tuned Qwen2.5 model still shows relatively low performance, yet the paper does not provide sufficient in-depth analysis to explain the possible reasons behind this result.
3. The connection between the two proposed tracks appears weak. The authors are encouraged to further justify the necessity of defining two separate tracks, as the current design may appear more like an engineering separation rather than a conceptual one.
4. It is unclear whether the authors have evaluated the quality of the collected dataset. Since the data come from 95 real users, real-world user interactions may include redundant or noisy operations. This raises the question of whether low data quality contributes to the generally poor agent performance observed in the experiments.

---

> ### Author Response · Authors · 2025-11-27
>
> Thank you for your feedback.
>
> 1. **Regarding the global generalizability:**
>
> Please refer to our response to the first question of Reviewer 7fJE.
>
> 2. **Regarding the fine-tuning:**
>
> We conducted more fine-tuning experiments on Qwen-2.5-VL-7B, including using the complete training set (16,000 episodes) and increasing the LoRA rank to 64. This will provide a much stronger evaluation of our data's potential. Please see Table 5 for the results.
>
> It can be seen that increasing the LoRA rank or the amount of training data both improve the model's performance, with the increase in training data having a particularly significant effect. When trained on the entire training set with a LoRA rank of 64, Qwen-2.5-VL-7B outperforms all the un-fine-tuned models in the experiment in terms of SR_1, Sim_1, and Sim_2, achieving the best performance. This indicates that the fine-tuned model on our data demonstrates stronger proactivity and personalization capabilities.
>
> 3. **Regarding the poor performance:**
>
> The poor performance demonstrates that our benchmark poses significant and non-trivial challenges that existing models find it difficult to solve. Existing models misinterpret subtle contextual information and historical intents in track 1 and default to a "generic" path instead of the user's preferred "personalized" path in track 2. However, after increasing the training data volume and the LoRA rank, the performance of the fine-tuned model has become excellent, surpassing all the un-fine-tuned models in terms of SR_1, Sim_1, and Sim_2. It is also worth noting that even with limited training data (1000 episodes), the performance of the fine-tuned model still outperforms large models such as GPT4.1. These results are crucial as they demonstrate the value of our user-oriented data. Fine-tuning on our benchmark can teach a model to better align with user-specific intent and action patterns.
>
> 4. **Regarding the online evaluation:**
>
> In personalized task execution, the focus of evaluation is not on the success rate of GUI control as existing benchmarks do, but on the agent's ability to execute tasks along a "personalized" path preferred by the user. Different users may take very different paths when performing similar tasks. The agent needs to imitate the user's preferences as a whole, interact with the environment from the beginning, plan and gradually build a personalized path. Therefore, we have chosen an online evaluation method that requires dynamic interaction with the environment and progressive task execution. This is also a more realistic evaluation method. If an offline evaluation method is adopted (that is, given the agent a fixed screenshot and goal, and requiring it to output a single step action), it mainly tests the agent's ability of GUI-grounding in a single step. The agent does not need to consider the overall path of the task, but only whether a single action is successful, thus failing to reflect the personalized path selection. Therefore, we believe that offline evaluation is not suitable for this benchmark. We plan to provide a simulator snapshot specifically for evaluation, including a fixed UI system, to enhance reproducibility and fairness.
>
> 5. **Regarding the connection between two tracks:**
>
> Due to the word limit, please see Appendix A.6.2 for complete response.
>
> 6. **Regarding the data quality:**
>
> To ensure the high quality of data, we did implement quality controls. We will further supplement these details in the appendix A.3:
>
> - User Training: Users were clearly informed during the training process that they should not perform redundant or useless operations during the demonstration, and the operation speed should not be too fast to avoid frequent repetitive operations. However, minor noisy operations (e.g., users making a typo or accidentally touching advertisements) are realistic situations in human interaction. A robust agent must be able to handle such scenarios. Even if the demonstrations are not collected from daily life but by recruiting annotators to perform operations in a simulator like existing datasets, such noise cannot be completely avoided. Therefore, we allow for its existence.
> - Manual Inspection: During the one-month data collection period, we conducted multiple timed quality checks on the data submitted by each user and manually deleted the low-quality data. We also provided quality feedback to the corresponding users, reminding them how to submit higher-quality data.
>
> As we mentioned in 3. Regarding the poor performance, the poor performance of existing models is a reflection of the task's difficulty, not low data quality. The excellent performance of our fine-tuned model demonstrates the value of our data.

---

### Official Review · Reviewer_7fJE · 2025-10-30

**Soundness:** 3
**Presentation:** 3
**Contribution:** 3
**Rating:** 6
**Confidence:** 4

**Summary:**

This paper introduces FingerTip 20K, a new benchmark for evaluating proactive task suggestion and personalized task execution in mobile LLM agents. The dataset consists of 21,437 real-world episodes gathered from 506 apps on 95 users’ daily Android device usage, with rich contextual meta annotations. The paper describes two evaluation tracks: (i) proactive intent suggestion drawing on multi-source user context, and (ii) personalized task completion that aligns with users’ habitual action trajectories. Extensive experiments with generalist and specialized models are presented, showing current systems’ limitations in leveraging user context, as well as initial improvements via fine-tuning on the new dataset.

**Strengths:**

1.	Real-World Data: Collected from long-term, in-the-wild mobile users, the dataset captures authentic intents and behaviors, offering far higher ecological validity than synthetic or simulator-based settings.
2.	Comprehensive Context Annotation: Each episode includes detailed user profiles, interaction metadata, and historical action, enabling advanced personalization and preference modeling.
3.	Rigorous Task & Metric Design: Both tracks are clearly formalized with well-defined metrics and baselines, providing a strong foundation for systematic evaluation.
4.	Open Access Commitment: Public release of data and code ensures reproducibility and promotes further research.

**Weaknesses:**

1.	Limited Demographic Scope: All data originate from Chinese Android users, restricting global generalizability and the benchmark may including implicit bias.
2.	Missing Related Work: The paper does not compare with several key studies on GUI benchmarks (e.g., AndroidWorld) and proactive LLM-based agents or personalization systems (e.g., AutoDroid, AppAgent).
3.	Superficial Ethics Discussion: Privacy and anonymization issues are acknowledged but insufficiently analyzed.

**Questions:**

1.	How do the authors envision adapting the benchmark and associated models to users or apps from non-Chinese regions? Are there plans for extending to other languages and UI ecosystems, and if so, how might the dataset, metrics, or methodology need to change?
2.	Are there specific plans for redacting or obfuscating sensitive visual/acoustic/UI data before public release? What concrete privacy-preserving steps are being implemented that go beyond user consent and manual filtering?
3.	Could the authors provide more detailed qualitative or quantitative analyses of how Sim_2 correlates with perceived personalization by real users? Are there user studies or human preference evaluations to complement the action-sequence similarity metrics, especially for subjective/latent aspects of personalization?

---

> ### Author Response · Authors · 2025-11-27
>
> Thank you for the detailed review.
>
> 1. **Regarding the global generalizability:**
>
> We acknowledge this limitation in Appendix A.1. Our primary goal for collecting the data was to capture deep and longitudinal user interactions in daily life settings. We believe that this context-rich dataset, even from a single region, provides a crucial foundation for the novel tasks of proactive task suggestion and personalized task execution. Considering the cost, we did not collect data in other regions. To our knowledge, previous datasets such as AitW and AndroidControl also contain a single language and UI ecosystem. We believe that this is a sufficient start for a first-of-its-kind study. However, we agree with the reviewer that user diversity is a crucial aspect in ensuring the global generalizability of our findings. To facilitate broader research, we plan to open source our APP for data collection. It can run on any (new version) Android personal phone, providing support for data collection in other regions and languages. We believe that our data collection methods and evaluation methods are universal.
>
> 2. **Regarding the related work:**
>
> Thank you for pointing this out. We have incorporated these works in Section 2 Related Work and included AndroidWorld in the comparison in Table 1. We have also added AutoDroid and AppAgent, two prompt engineering-based agents (using GPT4.1 as the base model), in Table 4 as baselines to test their personalized task execution capabilities. It should be noted that these two agents are designed for GUI control and, although they have the ability to proactively explore apps, they are not suitable for the proactive task suggestion we proposed. Please see Table 4 for the results. AppAgent achieved the best performance among all models in Sim_2 and Step Ratio, possibly due to its proficiency in learning from human demonstrations, but the time and token costs also significantly increased.
>
> 3. **Regarding the privacy:**
>
> We have discussed possible ethical issues in Appendices A.1, A.2, and A.3. We take data privacy very seriously and have taken multiple steps to mitigate this risk.
>
> - User Training: Participants were fully informed about the data usage, signed consent forms, and were explicitly instructed not to upload any data related to private information during training process. We provided participants with detailed guidance documents and video tutorials. We also provided them with targeted feedback on the data they submitted to ensure they understood which data could not be uploaded.
> - User Control: Participants can check or delete the data they upload at any time.
> - Human-machine hybrid inspection: We conducted two rounds of inspections. We first manually inspected the data and removed those that obviously involved privacy. Then, we used Qwen-VL-Max to examine the first and last screenshots of each episode and determine whether it involved privacy. Those episodes marked as potentially involving privacy were then rechecked by humans.
>
> We will further supplement these details in the appendix A.3. We acknowledge that more convenient automated inspection and editing technologies for privacy are crucial future directions.
>
> 4. **Regarding the validity of Sim_2:**
>
> This is a great question. Sim_2 is an automated metric for personalization. To quantitatively analyze the correlation between Sim_2 and users' subjective experience, we conducted a user study. Specifically, for the four models in Table 5, we combined their output (i.e., the complete action sequences output by the models) on the personalized task execution test set (200 episodes) with the users' true action sequences. Each episode has one ground truth action sequence and four randomly ordered action sequences output by the models. Then, we asked the users corresponding to these 200 episodes to rate the four models' action sequences on a five-point scale. The rating principle was whether the action sequence was personalized to execute the task according to the user's unique habits and preferences, even if it might not have been ultimately successful. Then, we calculated the average rating of the four models and compared it with their Sim_2. The results are as follows:
>
> |       Model       |  Sim_2   | User Rating |
> | :---------------: | :------: | :---------: |
> |  Qwen-2.5-VL-7B   |   0.95   |    2.42     |
> | Qwen-2.5-VL-7B-FT | **1.21** |  **3.35**   |
> |      GPT-4.1      |   0.98   |    2.55     |
> |  UI-TARS-1.5-7B   |   1.06   |    2.72     |
>
> It can be seen that the user rating increases with the increase of Sim_2, indicating a certain positive correlation between Sim_2 and users' subjective personalized experience. The fine-tuned model has the highest Sim_2 and its user rating also reached the highest 3.35 points, indicating that fine-tuning on our data indeed improved the model's personalization ability. We will supplement this part of the results in Appendix A.6.

---

### Official Review · Reviewer_mcYs · 2025-11-01

**Soundness:** 3
**Presentation:** 2
**Contribution:** 3
**Rating:** 4
**Confidence:** 3

**Summary:**

The paper presents a new benchmark to evaluate the proactive and personalized mobile llm agents, which is rarely explored in the literature. The new benchmark contains 20K unique human demonstrations of multi-step Android device interactions across a variety of everyday apps. Based on the proposed benchmark, a set of algorithms have been evaluated which can serve as the baselines for future research work. Also, a finetuned version based on the collected data can obviously improve the baselines.

**Strengths:**

* The new benchmark targets the twos problems of proactive task suggestion and personalized task execution, which are import to agent applications.

* The proposed benchmark is well designed with sufficient diversity covering many real-world applications.

* The paper reports the results of a set of algorithms which can be used as baselines for future research. Also, the proposed evaluation metric seems to be reasonable to correspond the human behaviors.

**Weaknesses:**

* The presentation of the paper should be improved. For example, the figures and charts in the paper can be replaced with vector image, which can provide better visual quality.

* For the baseline evaluations in Table 3 and Table 4, it would be better to include more vlms for a more complete evaluation. For example, how about the performance with more parameters like 72B vs 7B? Also, how about the performance with thinking model compared with non-thinking model?

* The user preference may be subjective and may vary depends on different situations. When the 20K data has been collected, is it possible to provide a probability setting for the human actions.

* In the paper, it claims there are 95 data collectors. How about the distribution of these collectors? Is it able to generalize the data collected in these users to other users? The generalization issue should be well considered otherwised the value of the benchmark may be challenged.

**Questions:**

Please mainly address the questions in the weakness section. More specifically, the main concern is on the limited baselines of the proposed benchmarks as well as the generalization ability based on the benchmark to other similar agent tasks.

---

> ### Author Response · Authors · 2025-11-27
>
> Thank you for your valuable feedback on improving our paper.
>
> 1. **Regarding the presentation of the paper:**
>
> We have replaced all figures with high-resolution vector graphics for maximum clarity.
>
> 2. **Regarding the baselines:**
>
> This is an excellent suggestion. For proactive task suggestion and personalized task execution, we have added two generalist models to the main experiments to provide a more complete set of baseline performances. Specifically, we introduced Qwen-2.5-VL-72B to compare with the existing 7B version; and Qwen-QVQ-Max (thinking model) to compare with other existing non-thinking models. Please see Table 3 and Table 4 for the results.
>
> It can be seen that after increasing the model parameters (from 7B to 72B), Qwen-2.5-VL achieved performance levels close to Qwen-VL-Max on both tracks. The thinking model Qwen-QVQ-Max surpassed GPT-4.1, achieving the new best performance among generalist models in both tracks, with success rates increasing to 12.8 and 9.5 respectively. This demonstrates the powerful capabilities of the thinking model. However, the time and token consumption of the thinking model also significantly increased.
>
> 3. **Regarding the probability setting:**
>
> This is an interesting idea. Our data is longitudinal, collected over one month. This means we often capture multiple instances of similar intents from the same user. This structure is precisely what allows for modeling user preferences and "habitual" intents. That is to say, within a specific time period of a day, a specific user's intents roughly follow a fixed probability distribution. We first separate all the intents of the same user by time periods (e.g., dividing a day into 24 time periods by hour). Then, we convert all the intents within the same time period into embedding vectors and cluster them based on distance. All semantically similar intents are regarded as one category. If the number of intents in a certain category is larger, it indicates that the probability of the user generating this type of intent during this time period is higher. In this way, we obtain the probability distribution of intents (e.g., the user has a 35% probability of ordering a hamburger for delivery and a 22% probability of playing music from a self-built playlist between 12:00 and 13:00...). For each user, a unique probability distribution of intents can be calculated through the above method.
>
> We re-executed the proactive task suggestion experiment by having GPT4.1 output the probability distribution of the user's intents instead of a single intent. The calculation method of SR_1 was changed to be successful as long as one of the top three intents in the output probability distribution could be regarded as the same as the user's true intent. The calculation method of Sim_1 was changed to the cosine similarity between the output probability distribution's embedding vector and the true probability distribution's embedding vector. It can be seen that by outputting the probability distribution, the agent provides multiple possible task suggestions, which is more likely to succeed than only outputting a single task suggestion. We will supplement this part of the results in Appendix A.6.
>
> |                    Setting                     | SR_1 | Sim_1 |
> | :--------------------------------------------: | :--: | :---: |
> |        Output a single intent directly         | 7.2  | 0.35  |
> | Output the probability distribution of intents | 11.1 | 0.42  |
>
> 4. **Regarding the generalization:**
>
> Users consist of one-third undergraduates, one-third postgraduates, and one-third employed individuals, including 54 males and 41 females, whose ages range from 18 to 60, with an average age of 25.9. We will provide these details on the user distribution in Appendix A.3.
>
> To explore the generalizability, we randomly sampled from the original test set and obtained three small test subsets, which are: (1) User-unseen, containing 126 episodes from 3 users. All data of these 3 users in the training set will be removed. (2) App-unseen, containing 106 episodes from 7 apps. All data of these 7 apps in the training set will be removed. (3) Intent-unseen, containing 99 episodes from 4 intent categories. All data of these 4 intent categories in the training set will be removed. The filtered training set has 14,706 episodes, and these data were used to re-fine-tune Qwen-2.5-VL-7B. The fine-tuned model was tested on these three out-of-domain test sets and the original test set. Please see Appendix A.6.1 for the results.
>
> It can be seen that when tested on new users, new apps, and new intent categories that have not been seen in the training set, the decline in model performance is not particularly severe. This indicates that the model fine-tuned on partial data has certain generalization ability and robustness, and can maintain good proactive task suggestion and personalized task execution capabilities in unseen data as well.

---

### Official Review · Reviewer_YFbi · 2025-11-01

**Soundness:** 3
**Presentation:** 3
**Contribution:** 4
**Rating:** 8
**Confidence:** 3

**Summary:**

The paper introduces FingerTip 20K, a mobile-agent benchmark built from real, in-the-wild phone usage: 95 users, ≈2 months, 21,437 episodes, 506 apps, avg. 11 steps per episode. Every episode is paired with user metadata (profile, time, place/context) and the user’s self-declared intent plus the full interaction trace (screens + a11y tree + actions). On top of this, the authors define two evaluation tracks that current GUI benchmarks don’t cover: (i) proactive task suggestion (given user + context + recent history, predict what the user likely wants to do now) and (ii) personalized task execution (given the task and this user’s past executions, complete it in this user’s style). Baselines with GPT-4-class models are far from human (≈7% vs 30% in proactive), showing the tasks are nontrivial.

**Strengths:**

- Truly real data. Unlike existing benchmarks that rely on emulator or auto-exploration, this is real phones + real users + real daily intents, so the distribution shift is authentic.
- New task formulations. Proactive recommendation and “execute like this user” are exactly what current mobile agents lack; existing benchmarks mostly test “can you follow a given instruction.”
- Rich context signals. Time, location category, user profile, and multi-intent history make it suitable for modeling preference and routine.
- Clear difficulty evidence. Even strong LLMs underperform humans → good headroom for the community.

**Weaknesses:**

- Privacy / deployment gap. Real-world agents won’t always have such clean, explicit user-intent annotations; some discussion of weaker supervision would help.

**Questions:**

See weakness

---

> ### Author Response · Authors · 2025-11-27
>
> Thank you for your excellent point about deployment. We agree that in future practical deployments, asking users to provide detailed descriptions for each of their intents would be cumbersome and may raise privacy concerns. Without requiring users to actively provide intent descriptions, the agent may need to make the following changes:
>
> From the perspective of agent input: In proactive task suggestion, the user's historical intent sequence is a necessary input. If the user's intent description cannot be directly obtained, one possible approach is to have the phone's accessibility feature automatically record the user's historical sequence at the action level (e.g., clicking the search box), and then introduce another agent to automatically summarize the high-level intent sequence from the low-level action sequence (e.g., all actions within a certain app over a period of time can be summarized into one intent). This ensures that the agent still has historical intent sequence as input. However, due to privacy concerns, users may also be reluctant to provide the action sequence. In this case, the agent can only make proactive task suggestion based on time, location, and user profiles. Our supplementary experiments show that when the user's historical intent sequence is removed from the input, GPT4.1's SR_1 drops from 7.2 to 0.8, and Sim_1 drops from 0.35 to 0.12, indicating a significant performance decline. This demonstrates that the historical sequence is crucial for proactive task suggestion (just as most current apps collect and analyze user behavior history). Therefore, a reasonable approach in future deployments is to strike a balance between automatically collecting user history sequences and protecting privacy.
>
> From the perspective of evaluation: In the actual deployment of evaluating whether the agent's proactive task suggestions are successful, if users do not provide real intents for comparison, they can simply give a binary judgment of acceptance/rejection or a numerical score for the agent's suggestions. The agent can receive feedback from these weaker forms of supervision and continue to learn.
>
> It should be noted that for the purpose of a benchmark, we argue that having clean, explicit, user-provided ground-truth intents is a necessity for reliable evaluation. Without this ground truth, it would be impossible to quantitatively measure an agent's success. In order to train models, we also need clear intents as labels. We frame our benchmark as the foundational ground-truth resource needed to tackle future work on weaker supervision.

---

### Author Response · Authors · 2025-12-01
**A summary comment**

Dear Area Chair,

We thank the reviewers for their constructive feedback and their recognition of our work's contribution: a **real-world, longitudinal dataset** (20k+ episodes, 95 users) and **two novel benchmark tracks** (proactive task suggestion & personalized task execution). To address the reviewers' concerns, we have conducted additional experiments during the rebuttal period. We highlight the key improvements below:

1. **Expanded Baselines (Addressing R-mcYs, R-7fJE)**

We introduced larger and more advanced models to the baselines:

- **Generalist models:** We evaluated **Qwen-2.5-VL-72B**, which showed performance gains over the 7B version. We evaluated a thinking model, **Qwen-QVQ-Max**, which achieved the best results among generalist models (SR_1: 12.8, SR_2: 9.5), surpassing GPT-4.1.
- **GUI agents based on prompt engineering:** We added **AutoDroid** and **AppAgent** as baselines for the personalized task execution track.

2. **Comprehensive Fine-Tuning (Addressing R-NkBD)**

We addressed the concern that the original fine-tuning (1k episodes) was too narrow by scaling up:

- We fine-tuned **Qwen-2.5-VL-7B on the full dataset (16,000 episodes)** with a higher LoRA rank (64). The fully fine-tuned model achieved **SR_1 of 26.0%, Sim_1 of 0.55, and Sim_2 of 1.42**, significantly outperforming all un-fine-tuned models and proving the dataset's efficacy in utilizing user-related information.

3. **Generalization & Metric Validation (Addressing R-mcYs, R-7fJE and R-NkBD)**

- **Robustness:** We created **"Unseen" test splits** (unseen users, apps, and intents). The fine-tuned model maintained robust performance on these splits, alleviating concerns about overfitting or lack of generalizability.
- **Global Access:** To address the regional limitation, we plan to open-sourcing our data collection App, enabling the community to extend this benchmark to other languages and regions.
- **Human Alignment:** We conducted a **user study** correlating our personalization metric (Sim_2) with human ratings. Results show a positive correlation (User Rating 3.35 for the model with highest Sim_2), validating our metric's effectiveness.

4. **Privacy and Presentation (Addressing R-mcYs, R-7fJE)**

- We have replaced all figures with high-resolution vector graphics.
- We have expanded the ethics discussion to detail our multi-step privacy protection.

We believe these revisions and new experiments strongly support the validity and value of FingerTip 20K as a foundational benchmark for the next generation of user-centric mobile agents. We believe that through detailed explanations, additional experiments, and further analysis, we have addressed all the major concerns. These modifications have been incorporated into the updated manuscript.

We would like to express our sincere gratitude once again to the reviewers for their insightful and constructive feedback, and also to the Area Chair for their valuable time and consideration spent on evaluating our work.

---

### Meta-Review · Area_Chair_ofPQ · 2025-12-16

**Summary:**

The paper introduces FingerTip 20K, a rich, real-world benchmark for evaluating proactive task suggestion and personalized task execution in mobile LLM agents. The dataset is notable for its scale (21,437 episodes), user diversity (95 real users), and breadth of app usage (506 apps), capturing authentic, longitudinal interactions. The benchmark is novel and well-motivated, addressing key gaps in current GUI agent evaluation.

**Reviewer Concerns:**

Reviewer YFbi’s concerns regarding deployment realism and reliance on explicit user intents are generally addressed in the rebuttal. The authors include additional experiments showing significant performance drops under weaker supervision.

Reviewer mcYs raises concerns about the limited set of baselines and generalization. These are addressed through new experiments with larger models, evaluating generalization to unseen users, apps, and intents, and improving the overall clarity and presentation of the paper.

Reviewer 7fJE requests more clarity on cross-regional generalization, related work, ethics, and metric validation. The rebuttal introduces a plan to open-source the data collection app, expands the discussion on ethical safeguards, and validates the Sim_2 personalization metric through a human evaluation study.

Reviewer NkBD raises multiple concerns about the fine-tuning setup, data quality, and evaluation design. The rebuttal responds with stronger fine-tuning results, explanations of evaluation choices, and detailed descriptions of data quality control procedures.

**Reviewer Scores:**

Based on the newly added experiments, I believe Reviewer mcYs would raise their score from 4 to 6. The other reviewers may maintain their initial scores. While the rebuttal is well-argued, some of Reviewer NkBD’s conceptual concerns remain partially unresolved.

---

### Decision · Program_Chairs · 2026-01-26

Accept (Poster)